# New Perspectives on the Risks of Hydroxylated Polychlorinated Biphenyl (OH-PCB) Exposure: Intestinal Flora α-Glucosidase Inhibition

**DOI:** 10.3390/toxics12040237

**Published:** 2024-03-24

**Authors:** Guoqiang Qin, Ruoyong Jia, Juntang Xue, Li Chen, Yang Li, Weiming Luo, Xiaomin Wu, Tianfeng An, Zhongze Fang

**Affiliations:** 1Department of Toxicology and Health Inspection and Quarantine, School of Public Health, Tianjin Medical University, Tianjin 300070, China; 2Department of Epidemiology and Biostatistics, School of Public Health, Tianjin Medical University, Tianjin 300070, China; 3National Center for Chronic and Noncommunicable Disease Control and Prevention, Chinese Center for Disease Control and Prevention, Beijing 100050, China; 4Tianjin Key Laboratory of Environment, Nutrition and Public Health, Tianjin 300070, China; 5National Demonstration Center for Experimental Preventive Medicine Education, Tianjin Medical University, Tianjin 300070, China; 6Tianjin Center for International Collaborative Research in Environment, Nutrition and Public Health, Tianjin 300070, China

**Keywords:** OH-PCBs, α-glucosidase, in vitro, inhibition kinetic parameters (Kis)

## Abstract

Polychlorinated biphenyls (PCBs) are a group of colorless and odorless environmental pollutants with a wide range of toxic effects. Some PCBs, especially less chlorinated ones, will rapidly undergo phase I metabolism after entering the body, and hydroxylated polychlorinated biphenyls (OH-PCBs) are the main metabolites of PCBs. Intestinal flora α-glucosidase is a common carbohydrate-active enzyme which is ubiquitous in human intestinal flora. It can convert complex dietary polysaccharides into monosaccharides, assisting the body in degrading complex carbohydrates and providing energy for the survival and growth of bacterial flora. The present study aims to investigate the inhibition of the activity of intestinal flora α-glucosidase by OH-PCBs. 4-Nitrophenyl-α-D-glucopyranoside (PNPG) was used as a probe substrate for α-glucosidase, and in vitro incubation experiments were conducted to study the inhibition of 26 representative OH-PCBs on α-glucosidase. Preliminary screening of in vitro incubation was performed with 100 μM of OH-PCBs. The results showed that 26 OH-PCBs generally exhibited strong inhibition of α-glucosidase. The concentration-dependent inhibition and half inhibition concentrations (IC50s) of OH-PCBs on α-glucosidase were determined. 4′-OH-PCB 86 and 4′-OH-PCB 106 were chosen as representative OH-PCBs, and the inhibition kinetic parameters (Kis) of inhibitors for α-glucosidase were determined. The inhibition kinetic parameters (Kis) of 4′-OH-PCB 86 and 4′-OH-PCB 106 for α-glucosidase are 1.007 μM and 0.538 μM, respectively. The silico docking method was used to further analyze the interaction mechanism between OH-PCBs and α-glucosidase. All these results will help us to understand the risks of OH-PCB exposure from a new perspective.

## 1. Introduction

Polychlorinated biphenyls (PCBs) are chlorine-containing organic compounds in which the hydrogen atoms in biphenyls are replaced by one or more chlorine atoms [1]. PCBs have stable physical and chemical properties, and can be ubiquitous in environmental substrates as well as in living organisms, with adverse effects on ecosystems and human health [2]. Like other persistent organic pollutants (POPs), PCBs can also accumulate in organisms and are amplified in the food chain due to their lipophilicity [3]. PCBs are mainly absorbed through the skin, respiratory tract and digestive tract, and are widely distributed in all the tissues of the body, of which fat tissue contains the highest PCB content [4]. As environmental pollutants, PCBs have a wide range of toxic effects. After being absorbed into the body, they can cause immunotoxicity, reproductive toxicity, neurotoxicity, teratogenicity and carcinogenicity. For example, exposure to PCBs is considered a risk factor for non-Hodgkin’s lymphoma (NHL), and immunosuppression may be one of the mechanisms that increase this risk [5]. It has been reported that exposure to PCBs may increase the risk of breast cancer, which may be related to the endocrine-disrupting mechanism of PCBs [6]. In addition, animal experiments have shown that PCBs can disrupt the control of neurotransmitters and thyroid hormones in the brain, thereby hindering the normal development of the brain and affecting intellectual development, encompassing aspects such as behavior, cognition and memory [7].

When PCBs enter the human body, oxidation reactions occur as a result of the catalysis of hepatic cytochrome P450 enzymes (CYP450) to generate hydroxylated polychlorinated biphenyls (OH-PCBs). Among the metabolites of PCBs, OH-PCBs are considered to be the most important. Previous studies have shown that approximately 40 different OH-PCBs are present in human blood, and these hydroxyl metabolites may be more biologically active and toxic than their parent compounds [8,9]. OH-PCBs are now recognized as a new environmental pollutant that can interfere with the endocrine function of the human body [10]. OH-PCBs have been reported to display a range of estrogenic and antiestrogenic activities [11]. In addition, studies have shown that certain OH-PCBs can cause thyroid dysfunction by competitively binding transthyretin (TTR) [12].

The human gut microbiota (HGM) is a complex community composed of numerous microorganisms, known as the “second brain” of the human body, which has an important impact on human health [13,14]. The intestinal flora develops along with our immune system to defend against invading pathogenic microorganisms and provide metabolic activities not encoded in the human genome [15]. Relevant studies have shown that enzymes encoded by intestinal flora can transform exogenous drugs, affect the pharmacokinetics and bioavailability of oral drugs, and jointly affect physiological balance with the body’s own enzyme system [16]. α-Glucosidase is a common carbohydrate-active enzyme, which is ubiquitous in the human intestinal flora. It can convert complex dietary polysaccharides into monosaccharides, provide energy for the survival and growth of flora, and assist the body in degrading complex dietary carbohydrates [17]. It has been reported that the active sites of human α-glucosidase and intestinal flora α-glucosidase have high sequence and structural identity, which indicates that most of the currently used α-glucosidase inhibitors can inhibit intestinal flora α-glucosidase [18]. When the activity of intestinal flora α-glucosidase is inhibited, delayed glucose absorption can be achieved, but this inhibition may also lead to fluctuations in the intestinal flora community [19].

The role of the gut microbiota in the body’s immune system has received growing attention, and POPs can further exert potential effects on host health and disease by affecting the physiological activities of intestinal flora [20,21]. Exposure to PCBs can affect gastrointestinal physiological function and the mucosal immune response, and alter the composition of intestinal flora [22]. As an important metabolite of PCBs, OH-PCBs have a certain toxic effect on the body. α-Glucosidase can provide energy for the survival and growth of intestinal flora and assist the body in degrading complex carbohydrates. Therefore, the purpose of this study is to investigate the inhibition of intestinal flora α-glucosidase by OH-PCBs. In this study, preliminary inhibition screening, concentration-dependent inhibition and inhibition kinetics were determined. In vitro–in vivo extrapolation (IVIVE) was used to predict the inhibition of α-glucosidase by OH-PCBs in vivo. Finally, in silico docking was used to help understand the mechanism of inhibition of α-glucosidase by OH-PCBs through a spatial structure simulation.

## 2. Materials and Methods

### 2.1. Chemicals and Reagents

Twenty-six OH-PCBs were purchased from J&K Chemical (Beijing, China). α-Glucosidase from *S. cerevisiae* (EC 3.2.1.20) and 4-Nitrophenol(PNP) were purchased from Sigma-Aldrich (St. Louis, MO, USA). The substrate 4-nitrophenyl-α-D-glucopyranoside (PNPG) and Na_2_CO_3_ were purchased from Macklin Biochemical Co., Ltd. (Shanghai, China). Phosphate buffer (Na_2_HPO_4_-NaH_2_PO_4_, PH6.8) was purchased from Leagene Biotechnology Co., Ltd. (Beijing, China). Ultra-performance liquid chromatography (UPLC)-grade acetonitrile was purchased from Tianjin Saifurui Technology Ltd. (Tianjin, China). Ultra-pure water was prepared by Millipore Elix 5 UV and Milli-Q Gradient Ultra-Pure Water System. All other reagents were of ultra-performance liquid chromatography (UPLC) grade or of the highest grade commercially available. 

### 2.2. In Vitro Incubation Experiment on α-Glucosidase and Enzyme Kinetics Study 

In this experiment, PNPG was used as substrate for α-glucosidase to determine the inhibition of α-glucosidase by OH-PCBs. The incubation system (total volume = 160 μL) contained OH-PCBs (0–100 μM), α-glucosidase (0.05 U/mL) and PNPG (250 μM); the concentration of phosphate buffer (Na_2_HPO_4_-NaH_2_PO_4_, PH6.8) used was 0.1 M. The incubation system containing OH-PCBs, α-glucosidase and phosphate buffer was pre-incubated at 37 °C for 10 min, and the substrate PNPG that was preheated for 10 min was added to the incubation system [18]. The reaction was carried out at 37.0 °C for 20 min. At the end of the reaction, 160 μL of Na_2_CO_3_ (1 M) was added for termination. The incubation mixture without OH-PCBs was used as control. The reaction system after constant temperature incubation was fully vortexed. After centrifugation at 12,000 rpm, 10 μL of supernatants was analyzed using an ultra-performance liquid chromatography (UPLC)-UV instrument. UPLC separation was achieved using a C18 column (100 × 2.1 mm, 1.7 μm). The flow rate of the mobile phase was set at 0.4 mL/min and the column temperature was 25 °C. The mobile phase was 0.5% formic acid aqueous solution for phase A and acetonitrile for phase B. The gradient conditions used were as follows: 0–7 min, 10% B; 7–9 min, 90% B; 9–16 min, 90% B; 16–19 min, 10% B. The detection wavelength of the system was 312 nm. 

In order to determine the Michaelis constant (Km) of the PNPG reaction catalyzed by α-glucosidase, the concentrations of PNPG for α-glucosidase were 62.5–625 μM. Incubation and analysis conditions were conducted as described above. The kinetic parameters Vmax and Km were determined by using Michaelis–Menten equation or the substrate inhibition equation via Prism 6 software.

### 2.3. Preliminary Screening of Inhibition Capability of OH-PCBs towards α-Glucosidase

To determine the inhibition of α-glucosidase by OH-PCBs, an incubation system (total volume = 160 μL) was used, containing OH-PCBs (100 μM), α-glucosidase (0.05 U/mL) and PNPG (250 μM); the concentration of phosphate buffer (Na_2_HPO_4_-NaH_2_PO_4_, PH6.8) used was 0.1 M. Other incubation conditions were conducted as described above. In addition, the incubation mixture without OH-PCBs was used as the negative control. All experiments were carried out in two independent experiments in duplicate. 

### 2.4. Half Inhibition Concentration (IC_50_) Determination and Inhibition Kinetics Evaluation

The concentration-dependent inhibition effect of different concentrations of OH-PCBs (ranging from 0 μM to 100 μM) on α-glucosidase was investigated, and the half inhibition concentration (IC_50_) was calculated. The inhibition kinetics were determined with multiple concentrations of PNPG (covering the Km value) and OH-PCBs (covering the IC_50_ values). Lineweaver–Burk plots were drawn using 1/reaction velocity (v) versus 1/the concentration of PNPG to determine the inhibition kinetics type. The second plot was drawn to determine the inhibition kinetics parameter (K_i_). For the second plot, the linear slope of the Lineweaver–Burk double reciprocal plot and the concentration of the inhibitor OH-PCBs were used.

### 2.5. In Vitro–In Vivo Extrapolation (IVIVE)

In vivo inhibition magnitude of α-glucosidase by OH-PCBs was determined through in vitro–in vivo extrapolation (IVIVE). The following equation was used:AUCi/AUC = 1 + [I]/Ki

The terms are defined as follows: AUCi/AUC was the predicted ratio of in vivo exposure of xenobiotics or endogenous substances with or without the co-exposure of OH-PCBs. [I] was the in vivo exposure concentration of OH-PCBs, and the K_i_ value was the in vitro inhibition constant. The standard was used as follows: [I]/K_i_ < 0.1, low possibility; 0.1 < [I]/K_i_ < 1, medium possibility; [I]/K_i_ > 1, high possibility.

### 2.6. In Silico Docking

In order to further understand the interaction mechanism between OH-PCBs and α-glucosidase, the in silico docking method was used to dock the chemical structure of OH-PCBs into the activity cavity of α-glucosidase. The three-dimensional (3D) structure of α-glucosidase was downloaded from the Protein Data Bank. Autodock software (version 4.2) was utilized to dock OH-PCBs into the activity cavity of α-glucosidase. The Lamarckian Genetic Algorithm (LGA) method was selected to perform the molecular docking study for the binding of OH-PCBs with α-glucosidase. The interactions between OH-PCBs and α-glucosidase were analyzed, including hydrogen bonds and hydrophobic contacts. 

### 2.7. Statistical Analysis of Data

The experimental data in this study are presented as the mean value plus standard deviation (S.D.). GraphPad Prism 5.0 was utilized to perform the statistical analysis. Statistical analysis between two groups was performed using a two-tailed unpaired Student’s *t*-test. Multiple groups were compared using a one-way ANOVA.

## 3. Results

### 3.1. Enzyme Kinetics Results and OH-PCBs Showed Broad Inhibition of α-Glucosidase 

PNP was detected by UPLC at 5.2 min, and the substrate PNPG was eluted at 3.5 min. The metabolism of α-glucosidase to PNPG conformed to the Michaelis–Menten equation (Appendix A). The Km value of PNPG catalyzed by α-glucosidase was 203.3 μM, and the Vmax value for the α-glucosidase-catalyzed PNPG metabolism was 1.13 umol/min/mg protein.

The inhibition of α-glucosidase by OH-PCBs is shown in Figure 1. A total of 100 μM of OH-PCBs showed strong inhibition of α-glucosidase. The inhibition rates of 4-OH-PCB1, 4-OH-PCB2, 4-OH-PCB14, 3′-OH-PCB9, 2′-OH-PCB9, 2′-OH-PCB5 and 2′-OH-PCB12 of α-glucosidase were 67.46%, 50.31%, 45.11%, 65.19%, 49.20%, 46.26% and 71.50%, respectively. And the inhibition rate of α-glucosidase of the other OH-PCBs was more than 90%. Through the preliminary screening results, we can find that OH-PCBs have a certain structure–activity relationship related to the inhibition of α-glucosidase. For example, the inhibition of OH-PCBs with three, four, five and six chlorine atoms on α-glucosidase was significantly stronger than that of OH-PCBs with one and two chlorine atoms.

### 3.2. Analysis of Inhibition Kinetics

4′-OH-PCB9, 4′-OH-PCB18, 2′-OH-PCB61, 3′-OH-PCB61, 4′-OH-PCB86 and 4′-OH-PCB106 were selected as the representative OH-PCBs, and the half inhibition concentrations (IC_50_) were determined. The concentration-dependent inhibition of α-glucosidase by 4′-OH-PCB9, 4′-OH-PCB18, 2′-OH-PCB61, 3′-OH-PCB61, 4′-OH-PCB86 and 4′-OH-PCB106 is given in Figure 2. The IC_50_ values for the inhibition of α-glucosidase by 4′-OH-PCB9, 4′-OH-PCB18, 2′-OH-PCB61, 3′-OH-PCB61, 4′-OH-PCB86 and 4′-OH-PCB106 were calculated to be 21.900, 18.260, 16.280, 5.664, 3.163 and 2.290 μM, respectively. Furthermore, the inhibition kinetics were determined. As shown in Figure 3A and Figure 4A, the intersection point was located on the vertical axis in the Lineweaver–Burk plot, indicating the competitive inhibition of α-glucosidase by 4′-OH-PCB86 and 4′-OH-PCB106. The second plots were drawn using the slopes of the lines in the Lineweaver–Burk plots as y versus the concentrations of inhibitors as x, and were used to calculate the inhibition kinetic parameters (Kis). Based on the second plots (Figure 3B and Figure 4B), the inhibition kinetic parameters (Kis) were calculated to be 1.007 μM and 0.538 μM for the inhibition of α-glucosidase by 4′-OH-PCB86 and 4′-OH-PCB106.

### 3.3. In Silico Docking to Elucidate the Inhibition Mechanism 

In silico docking was used to investigate the inhibition mechanism of OH-PCBs against α-glucosidase activity, and the binding free energy and active site for OH-PCBs on α-glucosidase can be obtained by this method. The representative docking results of 4′-OH-PCB9 and 2′-OH-PCB61 and α-glucosidase are given. As shown in Figure 5A, the active site where α-glucosidase binds to 4′-OH-PCB9 consists of the amino acid residues ASP-242, SER-240, LYS-156, LEU-313, SER-311, ARG-315, THR-310, ASP-307, PRO-312 and HIS-280. As shown in Figure 5D, the active site where α-glucosidase binds to 2′-OH-PCB61 consists of the amino acid residues HIS-280, ASP-242, SER-240, LYS-156, LEU-313 and PRO-312. 4′-OH-PCB9 and 2′-OH-PCB61 formed three and one hydrogen bonds with α-glucosidase amino acid residues, respectively (Figure 5B,E). As shown in Figure 5C,F, both 4′-OH-PCB9 and 2′-OH-PCB61 formed hydrophobic contacts with the amino acid residues Leu313 and Pro312 in the active cavity of α-glucosidase. In addition, the binding free energy of 4′-OH-PCB9 and 2′-OH-PCB61 with α-glucosidase were −6.56 and −6.06 kcal/mol, respectively. 

## 4. Discussion

Some PCBs, especially less chlorinated ones, are rapidly metabolized into OH-PCBs in vivo. As important metabolites of PCBs, OH-PCBs may have complicated influences on human metabolism. In this study, the inhibition of 26 OH-PCBs on intestinal flora α-glucosidase was investigated by constructing an in vitro determination system, and significant inhibition of the activity of α-glucosidase by OH-PCBs was demonstrated.

According to our results, a certain structure–activity relationship could be found. As we calculated, the IC_50_ values for the inhibition of α-glucosidase by two-chlorinated 4′-OH-PCB 9; three-chlorinated 4′-OH-PCB18; four-chlorinated 2′-OH-PCB61 and 3′-OH-PCB61; and five-chlorinated 4′-OH-PCB106 were 21.900, 18.260, 16.280, 5.664, 3.163 and 2.290 μM, respectively. Therefore, we found that the inhibition of α-glucosidase by OH-PCBs tends to be stronger with an increased amount of chlorine atoms. In addition, as calculated by the docking analysis, the binding free energy of 4′-OH-PCB9 and 2′-OH-PCB61 with α-glucosidase were −6.56 and −6.06 kcal/mol, which means 4′-OH-PCB9 could be more easily bonded to the enzyme. However, 4′-OH-PCB9 exhibited lower inhibitory ability against α-glucosidase, indicating that the binding free energy between OH-PCBs and α-glucosidase could not represent the inhibition potential. 

As one of the common carbohydrate-active enzymes, α-glucosidase is ubiquitous in the human intestinal flora, providing energy for the survival and growth of intestinal flora, and assisting the body in degrading complex carbohydrates [19,23]. Previous studies have shown that the inhibition of α-glucosidase and other carbohydrate-active enzymes may affect the carbohydrate metabolism of intestinal flora, which may lead to fluctuations in the intestinal flora community [24,25]. The human intestinal flora plays an important role in health, and its abundance and diversity are influenced by the dietary carbohydrate content [26]. The indigestible polysaccharides of the human body can be decomposed and fermented under the action of intestinal flora, and the disturbance of intestinal flora may be related to the occurrence and development of metabolic diseases [27]. The intestinal flora encodes a variety of enzymes that have the ability to bind and metabolize multiple bioactive compounds, such as transforming exogenous drugs, and affecting the pharmacokinetics and bioavailability of oral drugs [28]. In addition, the survival of intestinal flora also depends on the energy supply of complex carbohydrates, in which the intestinal flora’s α-glucosidase may play a significant role. Hence, it should be noted that when OH-PCBs enter the body and inhibit the activity of α-glucosidase, it may cause a disturbance in the intestinal flora, which is detrimental to human health. 

The potential inhibition of α-glucosidase by OH-PCBs in vivo was calculated through in vitro–in vivo extrapolation (IVIVE). 4′-OH-PCB86 and 4′-OH-PCB106 were selected as representative OH-PCBs. According to the [I]/Ki ratio ([I]/Ki > 0.1) evaluation standard, the threshold values were calculated to be 0.1007 μM and 0.0538 μM for the inhibition of α-glucosidase by 4′-OH-PCB86 and 4′-OH-PCB106, respectively. Therefore, when the exposure concentration of 4′-OH-PCB86 exceeds 0.1007 μM in vivo, the activity of α-glucosidase may be inhibited. Similarly, when the exposure concentration of 4′-OH-PCB106 exceeds 0.0538 μM in vivo, the activity of α-glucosidase may be inhibited. 

Our previous studies have demonstrated that OH-PCBs can inhibit the activity of UDP-glucuronosyltransferases (UGTs) and sulfotransferases (SULTs), thereby affecting the normal physiological functions mediated by enzymes [29,30]. The metabolic clearance of exogenous substances and endogenous small molecules in vivo requires a series of enzymatic reactions, among which phase II metabolic enzymes such as UGTs and SULTs play an important role. The intestinal flora has genes far beyond those contained in the human genome, encoding enzymes that provide metabolic activities not mediated in the human genome [31]. In this study, we found that OH-PCBs can inhibit the activity of α-glucosidase, which may affect the normal physiological metabolism of intestinal flora. The intestinal flora and human body are interdependent and develop together, and have a certain impact on health [32]. Therefore, when the concentration of OH-PCBs exposed in vivo exceeds a certain threshold and inhibits phase II metabolic enzymes such as UGTs and SULTs, as well as enzymes derived from intestinal flora such as α-glucosidase, it may cause more serious damage to the human body. 

## 5. Conclusions

This study fully described the inhibition of α-glucosidase by OH-PCBs, and determined the half inhibition concentration (IC50) values and inhibition kinetics. In vitro–in vivo extrapolation (IVIVE) was used to predict the inhibition threshold of 4′-OH-PCB86 and 4′-OH-PCB106 against α-glucosidase in vivo. In silico docking was used to investigate the mechanism of the inhibition of α-glucosidase of the OH-PCBs. All these results will provide a new perspective for the study of the toxicity of OH-PCBs.

## Figures and Tables

**Figure 1 toxics-12-00237-f001:**
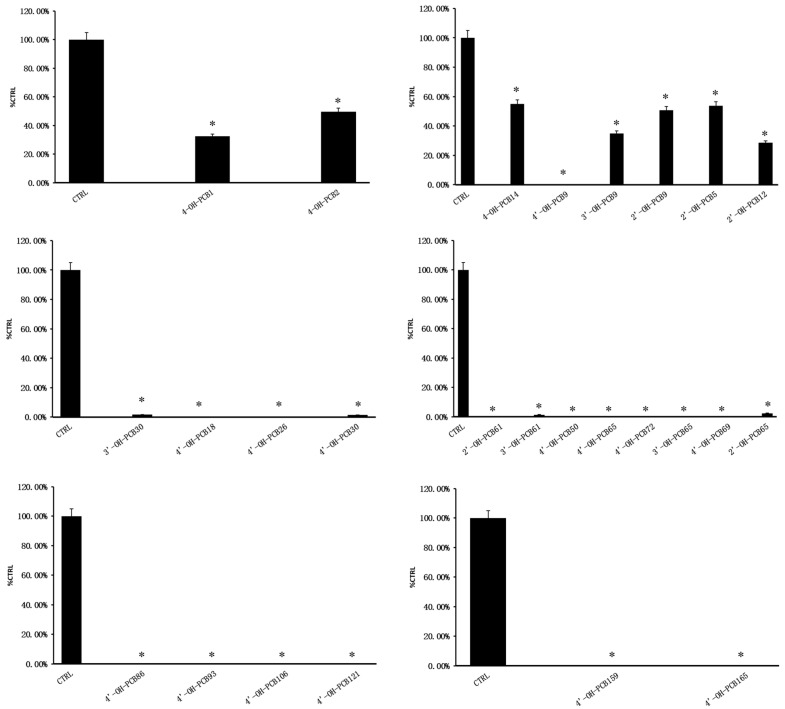
The preliminary inhibition screening of α-glucosidase by OH-PCBs. The data are given as mean value plus S.D., * *p* < 0.05.

**Figure 2 toxics-12-00237-f002:**
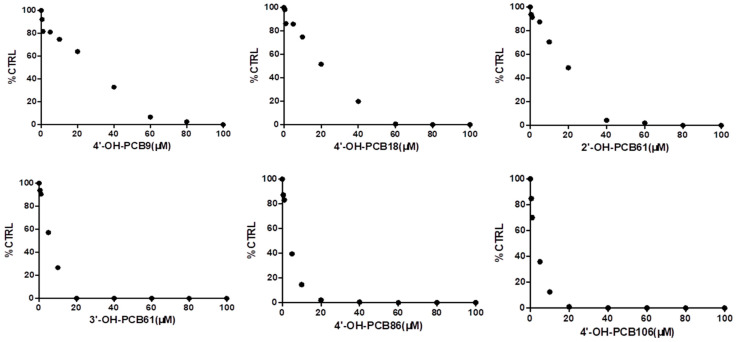
The concentration-dependent inhibition of α-glucosidase by 4′-OH-PCB9, 4′-OH-PCB18, 2′-OH-PCB61, 3′-OH-PCB61, 4′-OH-PCB86 and 4′-OH-PCB106. IC50 was determined by different concentrations of OH-PCBs. Parallel samples were taken, and the average values were used to draw the graph. Data are presented as the mean value plus S.D.

**Figure 3 toxics-12-00237-f003:**
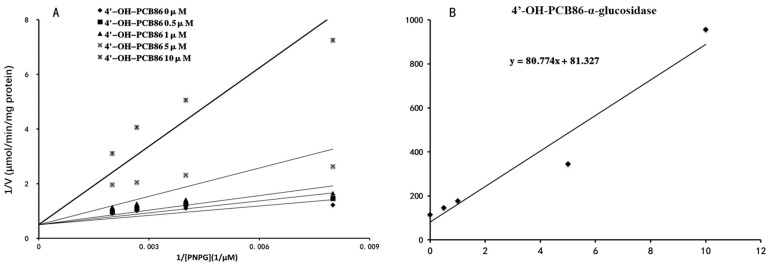
Inhibition kinetics of 4′-OH-PCB86 and α-glucosidase. Lineweaver–Burk plot of the inhibition the activity of α-glucosidase by 4′-OH-PCB86 (**A**). Each data point represents the mean value of duplicate experiments. Determination of inhibition kinetic parameter (Ki) of the inhibition of α-glucosidase activity by 4′-OH-PCB86 (**B**) using the second plots. The vertical axis represents the slopes of the lines from Lineweaver–Burk plots, and the horizontal axis represents the concentrations of 4′-OH-PCB86.

**Figure 4 toxics-12-00237-f004:**
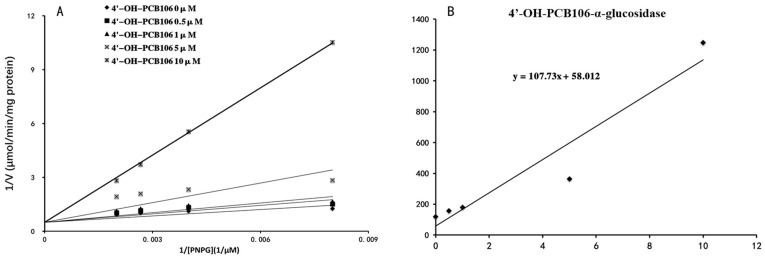
Inhibition kinetics of 4′-OH-PCB106 and α-glucosidase. Lineweaver–Burk plot of the inhibition of the activity of α-glucosidase by 4′-OH-PCB106 (**A**). Each data point represents the mean value of duplicate experiments. Determination of inhibition kinetic parameter (Ki) of the inhibition of α-glucosidase activity by 4′-OH-PCB106 (**B**) using the second plots. The vertical axis represents the slopes of the lines from Lineweaver–Burk plots, and the horizontal axis represents the concentrations of 4′-OH-PCB106.

**Figure 5 toxics-12-00237-f005:**
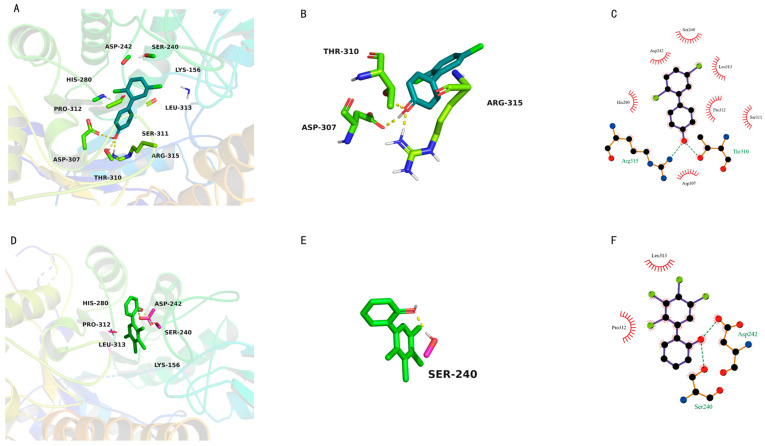
Active site of α-glucosidase binding with 4′-OH-PCB9 (**A**) and 2′-OH-PCB61 (**D**). Hydrogen bond interactions between 4′-OH-PCB9 (**B**) and 2′-OH-PCB61 (**E**) with the amino acid residues of α-glucosidase. Hydrophobic interaction between 4′-OH-PCB9 (**C**) and 2′-OH-PCB61 (**F**) and the active cavity of α-glucosidase.

## Data Availability

No new data were created or analyzed in this study. Data sharing is not applicable to this article.

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
