# Peer review of "New Perspectives on the Risks of Hydroxylated Polychlorinated Biphenyl (OH-PCB) Exposure: Intestinal Flora α-Glucosidase Inhibition"

_toxics, 2024, doi:10.3390/toxics12040237_

Round 1
Reviewer 1 Report
Comments and Suggestions for Authors
My comments are attached. My opinion is to accept the paper as it is.

Author Response
Comments 1: I can agree with the content of this paper and it can be published as it is and I advise to accept the paper as it is.
Response 1: Sincerely thanks for your examination and review of our manuscript, we were encouraged by the praises to our work and the positive comment.
Reviewer 2 Report
Comments and Suggestions for Authors
In the manuscript titled “New Perspectives on the Risks of Hydroxylated Polychlorinated Biphenyls (OH-PCBs) Exposure: Intestinal Flora α-Glucosidase Was Inhibited”, the authors have investigated the inhibition of some OH-PCBs to the flora alfa-glucosidase. The manuscript organization and writing are acceptable, and the experiment designs are reasonable and can support the statement. However, some concerns and comments are listed below.
1) Line 23-24, Line 257. Some statements about OH-PCBs are not accurate. There are 209 PCB congeners, some PCBs, especially lower chlorinated PCBs are easily metabolized to OH-PCBs, but some are not, like PCB126. So, the statement by the authors needs to be rewritten. Line 23-24, and Line 257, “PCBs will rapidly undergo phase I metabolism after entering the body …”, these statements are suggested to describe more accurately, for instance, “Some PCBs, especially lower chlorinated PCBs are…”.
2) Line 112, Line 114, Line 124, Line 145. There were some basic chemical formula issues. The formula of some chemicals is not correct. Line 112, “4-Nitrogen-α-D-glucopyranoside”, should be “4-nitrogen-α-D-glucopyranoside”. In “Na2HPO4-NaH2PO4” , the numbers should be in subscripts.
3) In Line 147, the authors stated that all experiments were carried out in two independently. It is generally accepted that all the experiments need to be performed at least triplet. Only two experiments would lower the confidence level of the values.
4) Since 26 OH-PCBs were investigated, further discussion about the OH-PCB structure, will make the manuscript more informative. For example, how the substitute position of OH- and Cl- affects the inhibitions.
Author Response
Comments 1: Line 23-24, Line 257. Some statements about OH-PCBs are not accurate. There are 209 PCB congeners, some PCBs, especially lower chlorinated PCBs are easily metabolized to OH-PCBs, but some are not, like PCB126. So, the statement by the authors needs to be rewritten. Line 23-24, and Line 257, “PCBs will rapidly undergo phase I metabolism after entering the body …”, these statements are suggested to describe more accurately, for instance, “Some PCBs, especially lower chlorinated PCBs are…”.
Response 1: Thank you for your suggestion. Based on your advice, sentences have been revised at Line 23-25, “Some PCBs, especially lower chlorinated ones will rapidly undergo phase Ⅰ metabolism after entering the body, and hydroxylated polychlorinated biphenyls (OH-PCBs) are the main metabolites of PCBs.” and at Line 262,“Some PCBs, especially lower chlorinated ones are rapidly metabolized into OH-PCBs in vivo.”
Comments 2: Line 112, Line 114, Line 124, Line 145. There were some basic chemical formula issues. The formula of some chemicals is not correct. Line 112, “4-Nitrogen-α-D-glucopyranoside”, should be “4-nitrogen-α-D-glucopyranoside”. In “Na2HPO4-NaH2PO4” , the numbers should be in subscripts.
Response 2: Thanks for your suggestion. We checked our manuscript again and revised the English expression of some formula. At Line 115, formula was revised as “4-nitrophenyl-α-D-glucopyranoside”. At Line 116, 126 and 148, formula was revised as “Na2HPO4-NaH2PO4”.
Comments 3: In Line 147, the authors stated that all experiments were carried out in two independently. It is generally accepted that all the experiments need to be performed at least triplet. Only two experiments would lower the confidence level of the values.
Response 3: Thank you for your suggestion. Certainly, common experiments need to be performed at least triplet. However, two reasons stopped us from this. Firstly, the cost of our research was really big, both OH-PCBs and enzymes cost much. And to give a more systematical test of as many OH-PCBs as possible, we tested 26 congeners commercially available for us. Therefore, we have to control our workload. Besides, standard deviation values were calculated to ensure that our results were credible enough. Secondly, we designed our experiments to be carried out independently in duplicate based on our former work and papers. Like in research of Sai-Nan Li in 2018 (Li SN, Cao YF, Sun XY, Yang K, Liang YJ, Gao SS, Fu ZW, Liu YZ, Yang K, Fang ZZ. Hydroxy metabolites of polychlorinated biphenyls (OH-PCBs) exhibit inhibitory effects on UDP-glucuronosyltransferases (UGTs). Chemosphere. 2018 Dec;212:513-522. doi: 10.1016/j.chemosphere.2018.08.040.), in research of Qiaoyun Yang in 2020 (Yang Q, Bai Y, Qin GQ, Jia RY, Zhu W, Zhang D, Fang ZZ. Inhibition of UDP-glucuronosyltransferases (UGTs) by polycyclic aromatic hydrocarbons (PAHs) and hydroxy-PAHs (OH-PAHs). Environ Pollut. 2020 Aug;263(Pt B):114521. doi: 10.1016/j.envpol.2020.114521.) and in research of Yong-zhe Liu in 2020 (Liu YZ, Pan LH, Bai Y, Yang K, Dong PP, Fang ZZ. Per- and polyfluoroalkyl substances exert strong inhibition towards human carboxylesterases. Environ Pollut. 2020 Aug;263(Pt A):114463. doi: 10.1016/j.envpol.2020.114463.).
Comments 4: Since 26 OH-PCBs were investigated, further discussion about the OH-PCB structure, will make the manuscript more informative. For example, how the substitute position of OH- and Cl- affects the inhibitions.
Response 4: Thanks for your advice, we revised part of our discussion at Line 268 “According to our results, a certain structure-activity relationship could be found. As we calculated, the IC50 values for the inhibition of 2 chlorinated 4'-OH-PCB 9, 3 chlorinated 4'-OH-PCB18, 4 chlorinated 2'-OH-PCB61 and 3'-OH-PCB61, 5 chlorinated 4'-OH-PCB106 towards α-glucosidase were 21.900, 18.260, 16.280, 5.664, 3.163 and 2.290μM, respectively. Therefore, we could found that the inhibition of OH-PCBs towards α-glucosidase tends to be stronger with the growth amount of chlorine atoms. Besides, as calculated by docking analysis, the binding free energy of 4'-OH-PCB9 and 2'-OH-PCB61 towards α-glucosidase were -6.56 and -6.06kcal/mol, which means 4'-OH-PCB9 could be more easily bonded to the enzyme. However, 4'-OH-PCB9 exhibited lower inhibitory ability towards α-glucosidase, indicating that the binding free energy between OH-PCBs and α-glucosidase could not represent the inhibition potential. ”.
About the relationship between positions of OH and Cl substituents and the inhibitory ability of OH-PCBs towards α-glucosidase, actually we tried to analyze through preliminary screening results and IC50 values. However, no stable rules were found, so we finally chose to not mention that.